# The cancer glycocalyx mediates intravascular adhesion and extravasation during metastatic dissemination

Giovanni S. Offeddu[1], Cynthia Hajal [2], Colleen R. Foley[1], Zhengpeng Wan[1], Lina Ibrahim [2], Mark F. Coughlin [1✉] & Roger D. Kamm [1,2✉]

The glycocalyx on tumor cells has been recently identified as an important driver for cancer progression, possibly providing critical opportunities for treatment. Metastasis, in particular, is often the limiting step in the survival to cancer, yet our understanding of how tumor cells escape the vascular system to initiate metastatic sites remains limited. Using an in vitro model of the human microvasculature, we assess here the importance of the tumor and vascular glycocalyces during tumor cell extravasation. Through selective manipulation of individual components of the glycocalyx, we reveal a mechanism whereby tumor cells prepare an adhesive vascular niche by depositing components of the glycocalyx along the endothelium. Accumulated hyaluronic acid shed by tumor cells subsequently mediates adhesion to the endothelium via the glycoprotein CD44. Trans-endothelial migration and invasion into the stroma occurs through binding of the isoform CD44v to components of the sub-endothelial extra-cellular matrix. Targeting of the hyaluronic acid-CD44 glycocalyx complex results in significant reduction in the extravasation of tumor cells. These studies provide evidence of tumor cells repurposing the glycocalyx to promote adhesive interactions leading to cancer progression. Such glycocalyx-mediated mechanisms may be therapeutically targeted to hinder metastasis and improve patient survival.

[1] Department of Biological Engineering, Massachusetts Institute of Technology, Cambridge, MA, USA. [2] Department of Mechanical Engineering, Massachusetts Institute of Technology, Cambridge, MA, USA. ✉email: mfcoughl@mit.edu; rdkamm@mit.edu

Tumor cells (TCs) across the metastatic cascade exhibit diverse strategies to circumvent physiological barriers that maintain homeostasis. In addition to genetic redirection and expression of increasingly malignant traits[1,2], disseminating TCs can highjack the normal function of platelets and immune cells like neutrophils to survive hemodynamic forces and immune clearance while traversing the blood circulation[3–5]. A growing body of evidence indicates that TCs can also modify distant microenvironments to facilitate survival and proliferation in the later stages of metastasis[6–8]. However, the formation of a vascular premetastatic niche, which may facilitate TC escape from the circulation[9,10], remains largely unexplored, and much is still unknown about how TCs achieve vascular egress (extravasation)[11,12]. As vascular dissemination represents the rate-limiting step in metastasis[13], uncovering mechanisms of interaction between TCs and endothelial cells (ECs) may reveal crucial therapeutic opportunities.

The extravasation of TCs includes two necessary and sequential steps: arrest under blood flow and transendothelial migration[14]. Both processes have largely been studied in terms of the molecules mediating specific adhesive interactions between TCs and ECs[15]. Remarkably, the EC adhesion molecules identified (e.g., selectins, ICAM-1, and VCAM-1[16–18]) are strongly associated with EC activation, bringing into question the requirement of preexisting inflammation for extravasation. Notably, cell adhesion is not only controlled through surface expression of specific receptors, but also by the glycocalyx (GCX), a ubiquitous surface coating of glycoproteins and proteoglycans. The GCX has traditionally been viewed as a repulsive barrier that prevents adhesion due to the dense, charged, and hydrated nature of its main components, such as hyaluronic acid (HA), heparan sulfate (HS), and chondroitin sulfate (CS)[19,20]. Curiously, the GCX on

TCs is particularly overexpressed and robust, yet it seems to enhance adhesion of TCs to both the normal and inflamed endothelium[21–23]. As such, the role of the GCX in TC adhesion and trans-endothelial migration is not yet resolved.

Direct observations of extravasation in vivo have been conducted in animal models, which may have limited relevance to human biology[24] and present intrinsic difficulties in visualization of rapid interactions between TCs and ECs. Therefore, in order to assess relevant TC extravasation mechanisms, advanced experimental systems are required to recapitulate human physiology while allowing for high spatiotemporal resolution imaging of cell interactions[14,25,26]. Our research group has developed and used perfusable microvascular networks (MVNs) self-assembled from human ECs and stromal cells within microfluidic devices[27], which have been employed in the past to explore the role of specific adhesion molecules, such as integrins, expressed on TCs[28]. The MVNs possess morphological similarities to the human microvasculature in vivo and express a functional GCX[29]. Perfusion of tumor cells in the MVNs can be used to quantify TC extravasation events[30]. Using MVNs, we dissect the process of TCs dissemination into arrest, adhesion, and *trans*-endothelial migration, and find that the GCX plays unexpected key roles in all of these processes ultimately leading to metastasis.

## Results

**Endothelial and tumor cell glycocalyces both determine extravasation efficiency.** We employed a well-established methodology developed by our group to quantify the extravasation efficiency of TCs in the MVNs[30]. MVNs were formed in microfluidic devices using primary human umbilical vein ECs (HUVEC) and normal human lung fibroblasts (nHLF) from

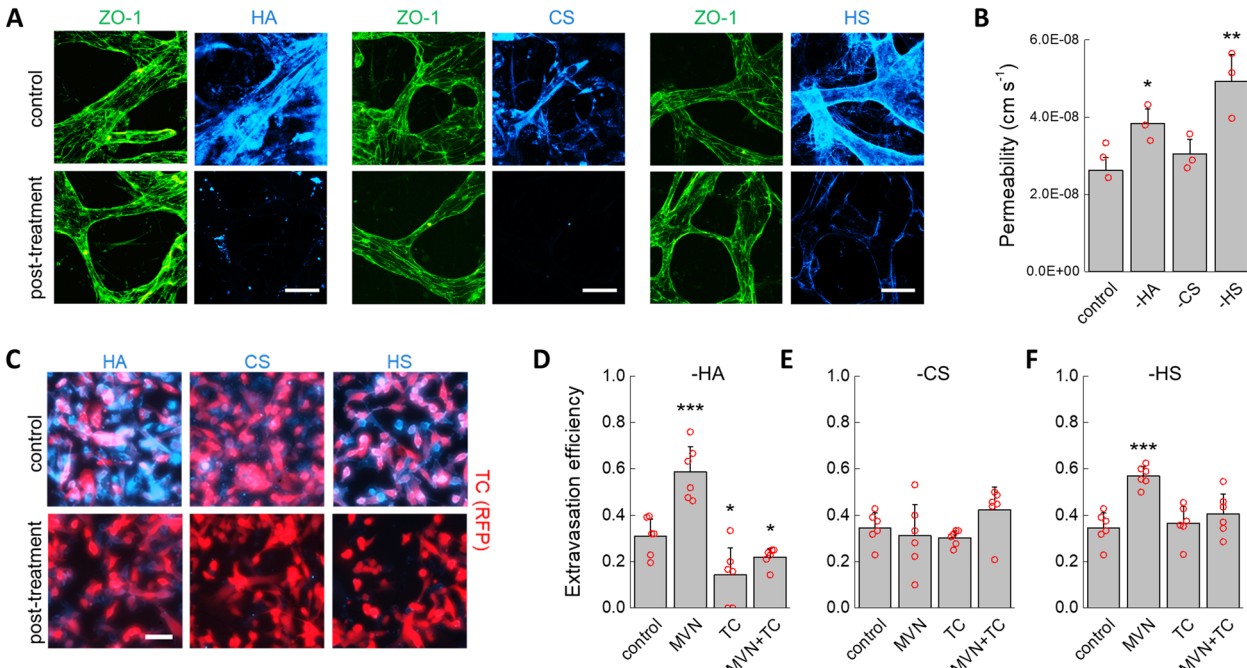

**Fig. 1 Effect of GCX degradation on extravasation of TCs. A** Fluorescent staining of endothelial junctions using ZO-1 and hyaluronic acid, chondroitin sulfate, and heparin sulfate before and after enzymatic removal from the microvascular network endothelium. **B** Changes in MVN permeability to 70 kDa dextran for untreated control networks and after enzymatic removal of HA, CS, and HS (*n* = 3 devices, three regions of interest averaged for each). **C** Immunofluorescent staining of HA, CS, and HS on TCs before and after enzymatic degradation. **D–F** Changes in extravasation efficiency of MDA-MB-231 cells after treatment of the MVNs, TCs, or both after enzymatic digestion of **D** hyaluronic acid, **E** chondroitin sulfate, and **F** heparin sulfate (*n* = 6 devices, 5 ROIs each). The scale bars in both (**A**) and (**C**) are 60 μm. Significance was assessed by student's *t* test assuming normally distributed data, *p* < 0.05 *, *p* < 0.001 ***.

pooled donors. Over 7 days, the ECs self-assembled into perfusable, interconnected vascular networks. In addition to recapitulating in vivo microvascular geometry and functionality as detailed elsewhere[29,31], the ECs formed an intact endothelium that expressed major components of the GCX: HA, CS, and HS (Fig. 1A). A 10 min perfusion of hyaluronidase, chondroitinase, or heparinase completely removed individual EC GCX components with little, if any, cross-reactivity (Fig. 1A; Supplementary Fig. 1). Targeted removal of HA and HS increased permeability of the MVNs to 70 kDa dextran by 1.4- and 1.7-fold compared to untreated control networks, respectively (Fig. 1B), consistent with the idea that the EC GCX can sterically hinder passage of macromolecules[32]. Indeed, this increase in permeability was not likely due to disrupted EC–cell junctions since ZO-1 localization revealed no visible changes in tight junction morphology, and treatment with the pro-inflammatory factor TNF-α, known to disrupt both the EC GCX and EC junctions[33,34], produced a much larger 6.9-fold increase in permeability (Supplementary Data). Thus, it is likely that the increase in permeability observed stemmed from increased diffusion of dextran across the degraded GCX layer between ECs through depletion of its solid, charged fraction, thereby increasing the GCX interfiber spacing and mean free path of the molecules[35,36]. Consistent with this mechanism, CS exhibited a much lower expression on the endothelium, and so its removal produced no measurable change in MVN permeability (Fig. 1B).

MDA-MB-231 breast cancer cells were chosen as model disseminating TCs due to their well-documented ability to rapidly extravasate from the MVNs[28,37]. Treatment of these cells with individual GCX-degrading enzymes resulted in the selective removal of HA, CS, and HS (Fig. 1C). Treated and untreated TCs were then perfused into the MVNs under a transient flow, and the number of extravasated and non-extravasated cells was counted after 6 h (Fig. 1D–F). Untreated TCs exhibited an extravasation efficiency of 33%. Selective removal of HA or HS from the MVN endothelium increased extravasation efficiency to 59% and 57%, respectively, while no change in extravasation efficiency was observed following removal of CS. These results suggest that expression of GCX components on the endothelium acts as a barrier to TC extravasation. Remarkably, removing HA from the TCs had the opposite effect of removing HA from ECs in that extravasation efficiency decreased by nearly half, to 14% (Fig. 1D). Removing the TC GCX has a stronger effect on extravasation efficiency than removing the MVN GCX, as depletion of HA from both MVNs and TCs resulted in a decrease in extravasation efficiency below control values at 22%. Removal of HS from TCs did not reduce extravasation efficiency below controls; however, it did counteract the increase in extravasation efficiency induced by removal of HS from the ECs (Fig. 1F). These observations show that the GCX has a profound effect on TC extravasation efficiency through a mechanism that goes beyond the GCX as a simple passive barrier that prevents cell–cell interactions. Since extravasation necessarily follows arrest, we next addressed the question as to whether changes in extravasation efficiency due to selective degradation of GCX components stem from changes in adhesive interactions upon arrest.

**Arrest of tumor cells in MVNs is determined by physical trapping and not specific adhesion.** To better understand if enzymatic digestion of the GCX modulates the initial arrest of TCs in the MVNs, we infused MDA-MB-231 cells into MVNs using a steady luminal flow and counted the fraction of cells that arrested (Fig. 2A). The device used for flow experiments had a 3 mm-wide gel channel[29,31,33] to enable formation of luminal paths reaching several millimeters, thus matching the length scale of the

microvasculature in vivo[38]. A constant pressure difference between 200 and 600 Pa gave rise to flow velocities, measured by fluorescent bead tracking, in the range 0.44–1.42 mm s$^{-1}$, which closely matches the 0.3–1 mm s$^{-1}$ physiological range[39]. Further, the fluid paths were entirely three-dimensional, as perfused MDA-MB-231 cells moved in and out of focus while frequently impacting the endothelium at vessel branch points (Supplementary Video 1 and Supplementary Fig. 2).

Arrest efficiency of MDA-MB-231 cells infused into the MVNs under a pressure difference of 200 Pa, i.e., at a mean flow speed of 0.44 mm s$^{-1}$, was high at 74% (Fig. 2B). Increasing the pressure difference to 400 and 600 Pa increased the luminal fluid speed to 0.92 and 1.42 mm s$^{-1}$, while reducing arrest efficiency to 32% and 22%, respectively. This result is consistent with previous observations indicating that local flow patterns and, consequently, shear stresses, are a key determinant of TC arrest[40]. Remarkably, TCs arrested in the MVNs by the two distinct mechanisms previously observed in vivo[41]: physical trapping and adhesive capture (Fig. 2C). Physical trapping was observed in smaller capillaries where TCs lodged between two or more surfaces of the endothelium. The less common adhesive capture was observed in larger vessels and lower flow speeds, where TCs appeared to be adhered to only one endothelial surface.

Specific degradation of HA, CS, or HS from ECs produced no change in average MVN vessel diameter (Fig. 2D). However, at a pressure difference of 400 Pa, used hereafter, degradation of GCX components increased average fluid speed compared to control values. This phenomenon has been observed in vivo[42], and it likely stems from an increase in the effective luminal diameter through a reduction in the thickness of the GCX, possibly larger than 1 μm[20,43], or perhaps an increase in compliance and flow-induced deformation of the GCX after depleting part of its charged solid fraction[44,45]. The increase in fluid speed with GCX degradation was comparable to increasing the driving pressure from 400 to 600 Pa, but instead of reducing arrest efficiency, removal of HA and HS from the MVNs increased arrest efficiency from 32% to 54% and 55%, respectively (Fig. 2E). The increase in arrest efficiency was specific to HA and HS on ECs: removing CS from ECs or removing any component of the GCX on TCs had no effect on arrest efficiency (Fig. 2E). Interestingly, the increase in arrest efficiency with removal of HA and HS from the MVNs was obliterated by concurrent removal of the same components from the TCs.

We further examined the different mechanisms of arrest after targeted HA degradation. When HA on both the ECs and TCs was present, 85% of TCs arrested by physical trapping in small capillaries (Fig. 2F). As this is the predominant mechanism for TC arrest in vivo[41], this result alone highlights the importance of a physiological vascular morphology to recapitulate critical cancer dissemination events. Removing HA from the MVN ECs had a profound effect on adhesive interactions by more than doubling the average fraction of cells arrested by adhesive capture from 15 to 33%. Removing HA from either TCs or both MVNs and TCs, instead, reduced specific adhesion from 15% to 9% and 8%, respectively. Critically, these changes in arrest were primarily determined by the GCX of the TCs since removing HA on both TCs and ECs resulted in rates of trapping and adhesion that match the values found for only removing the GCX of TCs.

Taken together, the arrest efficiency and arrest mechanism data indicate that removal of HA from ECs slightly increased the number of trapped cells but nearly tripled the number of adhered cells. Thus, the role of the MVN GCX seems to be similar for TCs and macromolecules: it inhibits access to the endothelium by providing an additional physical barrier. However, the role of the TC GCX, particularly HA, is more subtle, as it appears to enhance adhesion. The next question we addressed is whether these changes in adhesion can stem from endothelial activation.

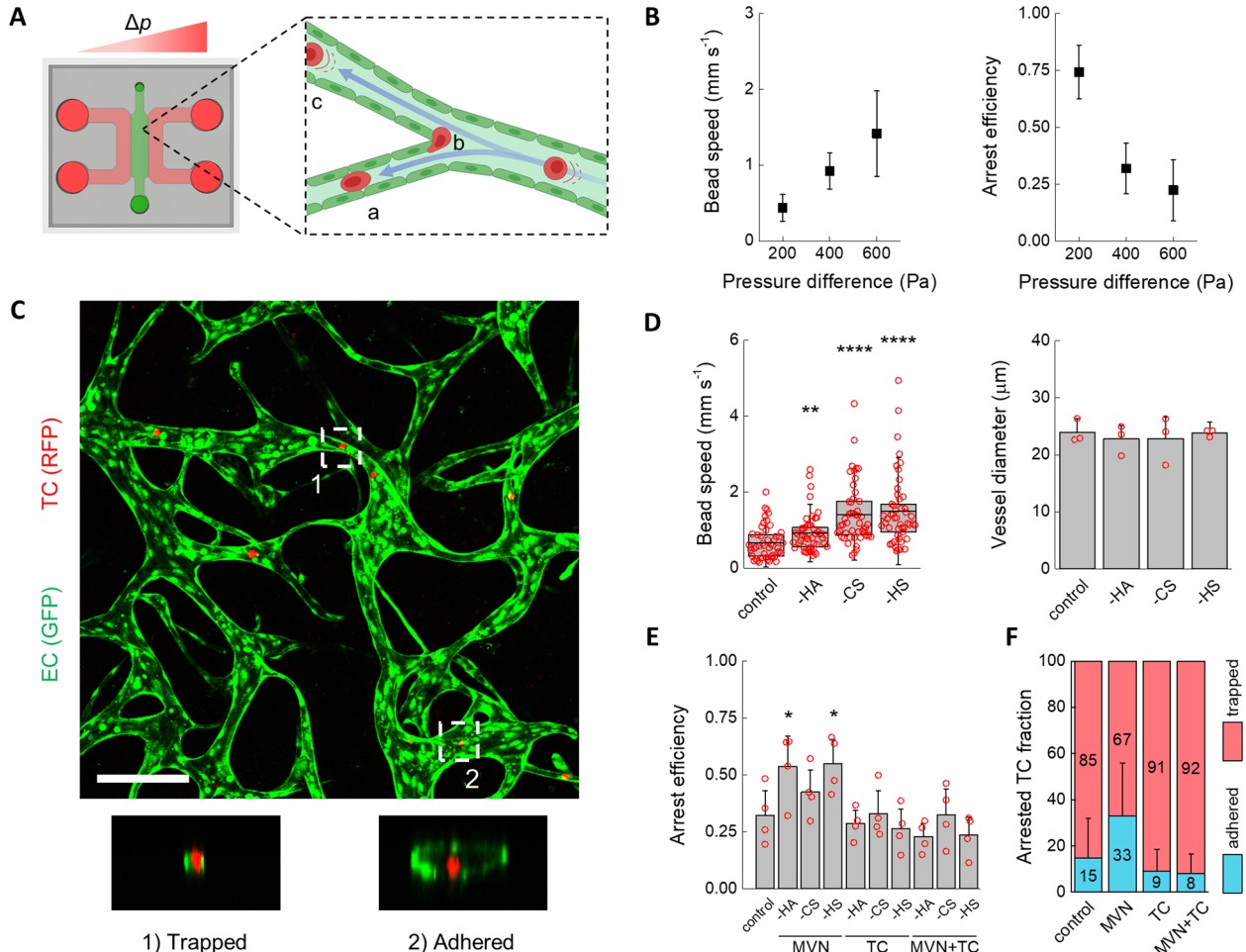

**Fig. 2 Characterization of TC arrest under flow in the MVNs. A** Schematic diagram of the microfluidic device used to perfuse MVNs under a constant pressure difference $\Delta p$ and cells being carried by luminal flow into **A** narrow channels and **B** impacting the endothelium at bifurcations or **C** large vessels (partially realized with Biorender.com). **B** Speed of inert beads carried by luminal flow driven by different pressure differences ($n = 50$ beads tracked in three devices, average and standard deviation shown) and arrest efficiency of TCs ($n = 4$ devices, average and standard deviation shown) as a function of the pressure difference. **C** Confocal image of two arrest mechanisms showing MDA-MB-231 cells within MVNs arrested by (1) physical trapping and (2) adhesion. The scale bar is 200 μm. **D** Bead speed through the MVNs ($n$ as above, the error bars indicate the standard deviation) and vessel diameter as a function of specific GCX component removal (the error bars indicate the standard deviation between the averages of $n = 3$ devices, 3 regions of interest each). **E** Arrest efficiency of TCs as a function of GCX enzymatic treatment of the MVNs alone, TCs alone, or both MVNs and TCs ($n = 4$ devices). **F** Cumulative percentage of TCs either physically trapped in small vessels or adhered to large vessels in the MVNs ($n > 40$ cells). Statistical significance was assessed by student's $t$ test assuming normally distributed data, $p < 0.05$ *, $p < 0.01$ **, $p < 0.0001$ ****. A normal distribution of the data in (**B**) and (**D**) was confirmed by Kolmogorov–Smirnov test.

**Endothelial activation is not necessary for tumor cell arrest and extravasation.** Our observations show that the GCX on the microvascular endothelium is a key determinant of TC adhesion and trapping. TCs are known to release pro-inflammatory cytokines which could locally regulate the EC GCX[16]. In addition, enzymatic removal of the GCX has been hypothesized to induce EC activation per se[46]. Therefore, we next sought to characterize the influence of GCX enzymatic treatment and adherent TCs on the activation state of the MVN ECs.

A high-sensitivity, semiquantitative automatic western blot assay was used to measure relative protein expression of adhesion molecules associated with EC activation. Treatment of the MVNs with GCX-degrading enzymes was carried out as before, and protein expression was analyzed after the same time as the TC extravasation experiments (6 h). Expression levels of receptors E-selectin, P-selectin, and VCAM-1 were low and did not change after enzymatic GCX removal (Fig. 3A). Instead, the pro-inflammatory cytokine TNF-α, used as positive control, produced

a marked increase in VCAM-1. A transient increase in E-selectin expression may be expected after challenging of the ECs[47], yet the lack of change in expression of the other proteins confirms that the enzymatic treatments did not induce EC activation. ICAM-1 expression increased following HA degradation to levels comparable to TNF-α exposure. However, this increase in ICAM-1 protein expression was mostly attributed to the stromal cells (Fig. 3B), as no quantitative or qualitative difference in expression levels between untreated and hyaluronase-treated ECs was observed by western blot (Fig. 3B) or confocal microscopy (Fig. 3C).

To determine if the presence of cancer cells arrested in the MVNs could induce local activation of ECs, we used high-resolution confocal microscopy to visualize ICAM-1 in the vicinity of arrested MDA-MB-231 cells after 6 h. Only a very marginal and sporadic increase in staining for ICAM-1 was observed in the vicinity of arrested TCs (Fig. 3D). It is possible that TCs other than MDA-MB-231 cells may induce EC

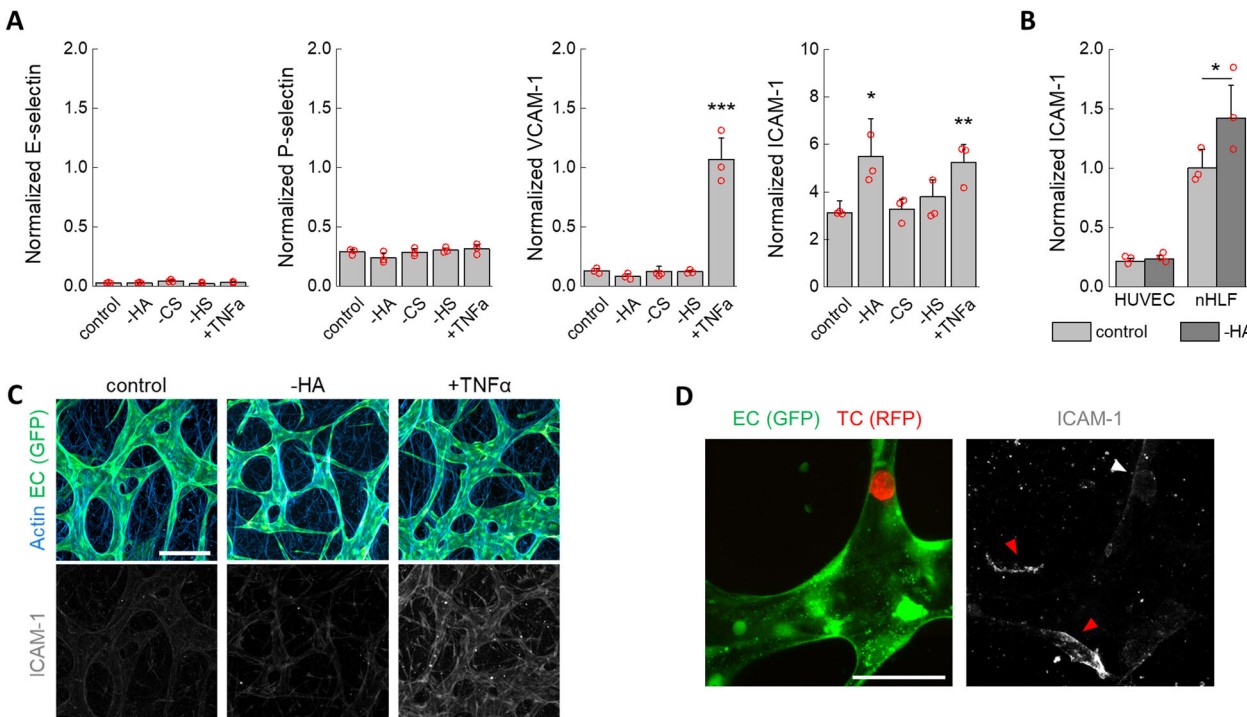

**Fig. 3 Absence of endothelial activation following GCX treatment. A** Normalized (to CD31) expression of EC adhesion molecules associated with endothelial activation ($n = 3$ devices). **B** Normalized (to β-actin) expression of ICAM-1 on ECs (HUVEC) and stromal cells (nHLF) cultured in well plates after treatment with hyaluronidase ($n = 3$ wells). **C** Imaging of ICAM-1 after MVN treatment. The scale bar is 200 μm. **D** ICAM-1 is not highly expressed on ECs in the vicinity of TCs arrested for 6 h prior to fixing (white arrow), as compared to ICAM-1 expressed on stromal cells (red arrows). The scale bar is 60 μm. Statistical significance was assessed by student's $t$ test assuming normally distributed data, $p < 0.05$ *, $p < 0.01$ **, $p < 0.001$ ***.

activation before extravasation, either through expression of different cytokines or by higher local cytokine concentrations through longer permanence in blood vessels. Nevertheless, our results show that TC-induced expression of EC receptors that participate in leukocyte trafficking are not strictly required for TC arrest and extravasation. Thus, we next sought to understand what alternative modes of adhesion to the endothelium may be used by TCs during extravasation.

**HA deposition by tumor cells creates a pro-metastatic vascular niche.** Observations in vivo show that TC arrest is often followed by intravascular motility, indicating adhesion of TCs to the endothelium[14]. In the MVNs, we sometimes observed TC migration within vessels after arrest (Supplementary Video 2). In addition, those TCs that eventually extravasated were, on average, more stationary under transient flow than those that did not extravasate (Supplementary Video 2, Supplementary Fig. 3a). Both behaviors suggest that adhesion between arrested TCs and microvascular ECs precede extravasation. Based on our results so far, we next sought to understand the role of HA in this interaction.

The GCX glycoprotein CD44 acts as the main receptor for HA and is often highly expressed by TCs, mediating adhesion to several components of the extracellular matrix (ECM)[48]. Here, CD44 was expressed by the MDA-MB-231 cells and localized at contact points between arrested TCs and the endothelium (Fig. 4A). CD44 anchors formed quickly, within several minutes, and they were also observed in small capillaries, i.e., between ECs and physically trapped TCs. Taken together, these observations suggest that rapid formation of CD44 anchors provides intravascular adhesion to resist flow after TC arrest. Interestingly, CD44 was also observed to localize between TCs (Fig. 4B) as recently observed in vivo for other breast TCs[49], which is

particularly important given the possibly higher metastatic potential of TC aggregates compared to single cells[50].

Regulation of arrest and extravasation efficiency has been attributed to a mechanism where CD44 expressed on EC interacts with HA on TCs[51]. However, CD44 is not typically present on ECs[52], except in some cancers[53], and expression levels of CD44 on the MVN ECs is low (Fig. 4A, B). An alternative mechanism has been proposed in which CD44 on TCs binds to fibronectin deposited on ECs under flow[54]. Yet, our data did not support this mechanism, whereby fibronectin was only localized on the basal surface of the endothelium (Supplementary Fig. 3b). Instead, we observed HA deposited upstream of TCs adherent to the MVNs via CD44 anchors. These streaks of HA are consistent with the interpretation that HA is transferred from TCs to ECs following contact and movement along the endothelium under flow. Indeed, HA streaks were observed in MVNs stripped of HA (Fig. 4B) as well as in untreated samples (Fig. 4C). Surprisingly, HA streaks extended past the arrested TCs (Fig. 4C, Supplementary Fig. 3c) possibly suggesting that HA may have been deposited by TCs that previously traversed those vessels without arrest.

This mechanism of local deposition of TC-bound HA in the microvasculature, which primes an adhesive niche, is shown in Fig. 4D. CD44 and HA on TCs play a role in TC trapping and adhesion in both small and large capillaries. That is, arrest efficiency is controlled by the TCs irrespective of the arrest mechanism. In this framework, the higher capacity of TCs to arrest in large capillaries after removal of EC-bound HA would be due to easier deposition of TC-bound HA on the bare endothelium in the absence of effective lubrication, i.e., repulsion, by the EC GCX. Both TC-bound HA and CD44 are needed for effective adhesion, as removal of HA from TCs increased their CD44 expression (Fig. 4E), yet fewer TCs arrested in the MVNs

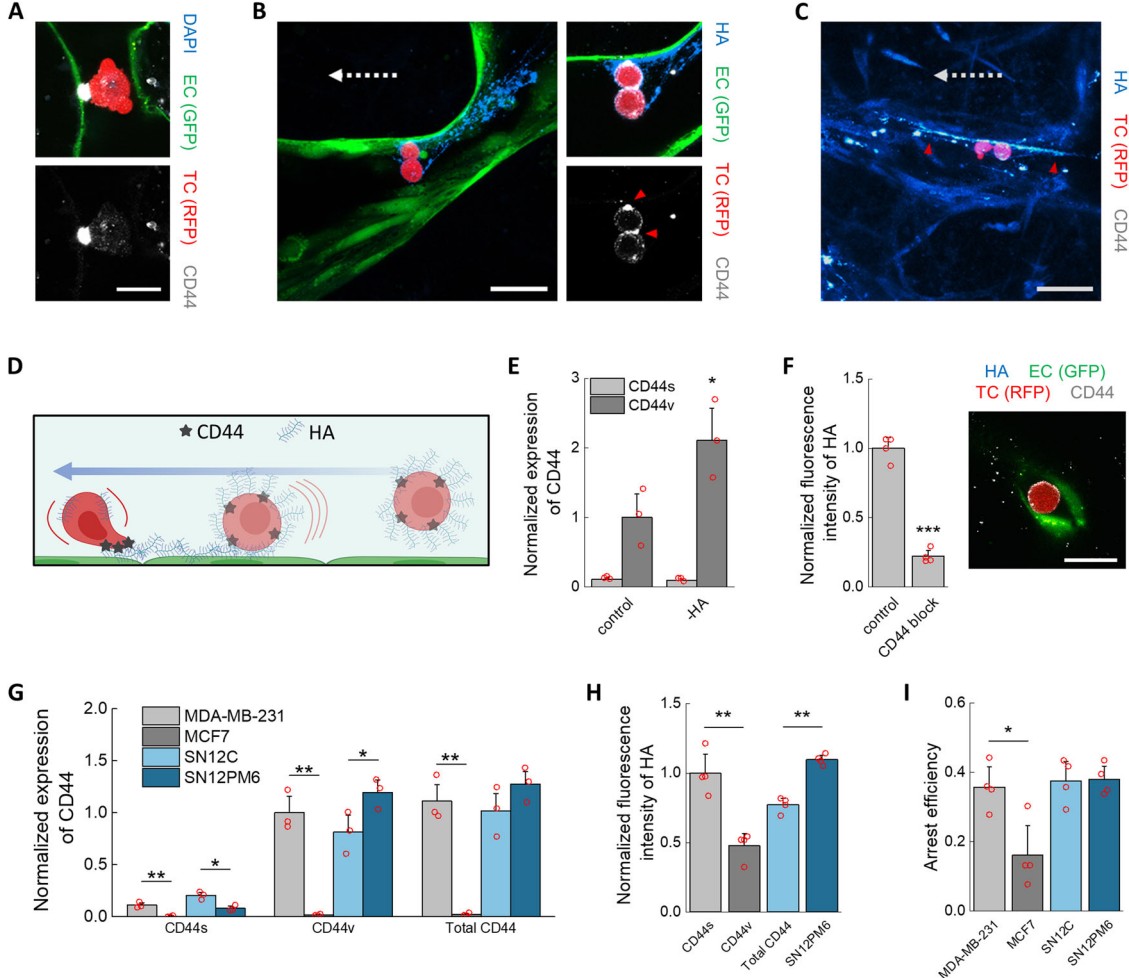

**Fig. 4 Mechanisms of arrest and adhesion mediated by the HA–CD44 complex. A** Confocal imaging of arrested TC in the microvasculature and CD44 anchoring. The scale bar is 20 μm. In (**B**), the TCs are bound through CD44 (red arrows) to streaks of HA. In **C**, the streaks (red arrows) continue past the TCs, possibly indicating previous deposition. The scale bars for both (**B**) and (**C**) are 60 μm. The dashed arrows indicate the direction of flow. **D** Diagram of GCX-mediated arrest mechanism of TCs, involving deposition of HA upon impact with the endothelium under flow and anchoring to HA through CD44 (partially realized with Biorender.com). **E** Normalized (to β-actin) expression of CD44 isoforms in TCs after treatment with hyaluronidase in well plates ($n = 3$ wells). **F** Expression of HA on TCs after antibody-blocking of CD44 ($n = 4$ wells) and confocal image of treated TC arrested in a small MVNs capillary, not showing CD44 anchoring or HA. The scale bar is 20 μm. **G** Normalized (to total protein concentration) expression of CD44 isoforms in various TCs, and **H** HA expression and **I** arrest efficiency in the MVNs of the same TCs ($n$ as above, shown for all). Statistical significance was assessed by student's $t$ test assuming normally distributed data, $p < 0.05$ *, $p < 0.01$ **, $p < 0.001$ ***.

through adhesion in large capillaries (Fig. 2). Remarkably, the increase in CD44 was only observed in the variant isoform of the glycoprotein (CD44v), and not in the standard isoform (CD44s). In addition, TCs treated with a CD44-blocking antibody showed shedding of HA and its local deposition could not be observed in the MVNs, while CD44 failed to form anchoring points (Fig. 4F, Supplementary Fig. 4).

To determine if CD44 and HA expression correlate with arrest efficiency, we measured the fraction of arrested cells in the MVNs for several well-characterized TC lines: the less-invasive breast cancer line MCF7, the weakly metastatic renal carcinoma cell line SN12C, and the strongly metastatic renal carcinoma subline SN12PM6[55]. We first determined that all cell lines showed more pronounced expression of CD44v than CD44s (Fig. 4G). Of the breast cancer lines, MDA-MB-231 cells express more CD44s and CD44v than MCF7 cells, which express by far the least of all four TCs. Of the renal carcinoma lines, SN12PM6 cells express more CD44v, while SN12C express more CD44s, with the two TCs expressing equal levels of CD44 overall. The expression of HA followed the trend in expression of CD44v and total CD44, with

MCF7 cells having the least amount of HA (Fig. 4H, Supplementary Fig. 4). Consistent with these findings, arrest efficiency in the MVNs was lowest in MCF7 cells, while the other three cell lines produced similar results (Fig. 4I), further confirming the importance of CD44–HA interactions in TCs for stable arrest and adhesion. Next, we sought to understand how these interactions can affect extravasation following arrest.

**CD44v, and not CD44s, mediates extravasation of tumor cells.** Binding of TC CD44 to HA deposited on vascular ECs induced arrest and adhesion of various TCs. At the same time, our results show that extravasation of MDA-MB-231 cells after 6 h is also regulated by HA on TCs (Fig. 1), thus possibly implicating CD44. Therefore, we next visualized the role of CD44 in TC extravasation in real time. For this purpose, we created an MDA-MB-231 line overexpressing fluorescent CD44. Live-imaging of extravasation through confocal microscopy (Fig. 5A, Supplementary Videos 3 and 4) showed that CD44 binds to the ECM throughout the extravasation process: During adhesion to the apical vascular endothelium, TCs extend protrusions between EC

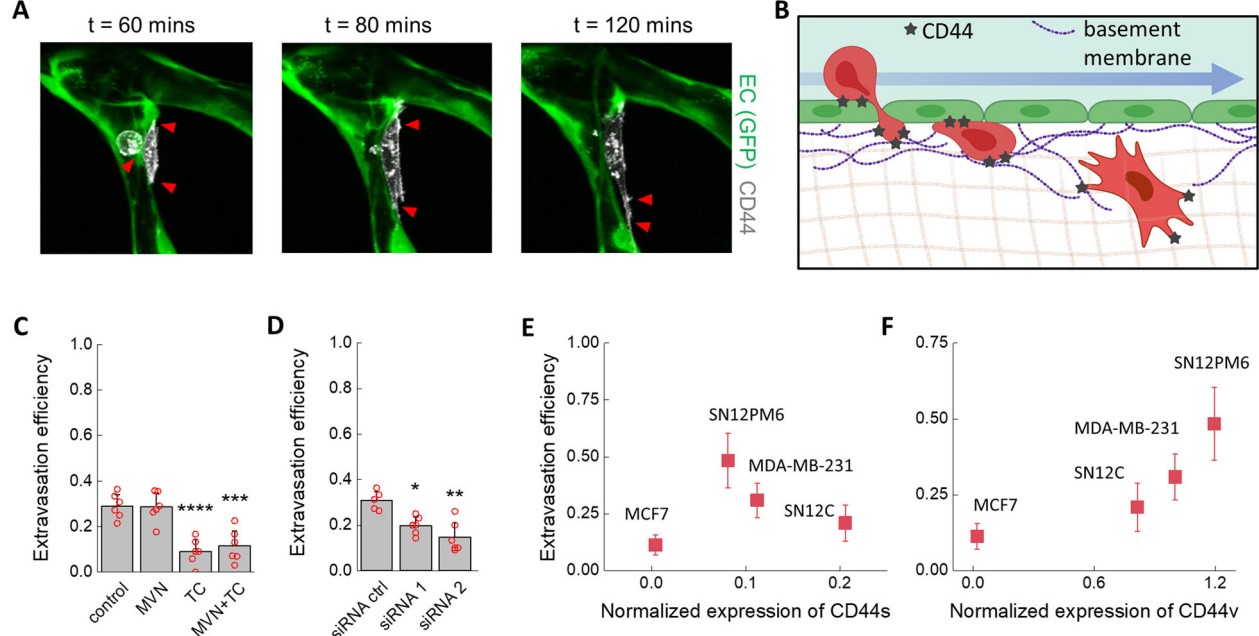

**Fig. 5 Role of CD44 during TC extravasation. A** Time-lapse confocal live imaging (scale bar = 20 μm, post-TC perfusion time) of extravasating TC showing the ECM-binding role of CD44 (red arrows) in the process, which includes protrusion through the endothelium, permanence in the basement membrane, and migration into the interstitium, schematically represented in (**B**), partially realized with Biorender.com. **C** Extravasation efficiency at 6 h after treatment with a CD44-blocking antibody ($n = 6$ devices) of TCs, MVNs, or both and **D** extravasation efficiency after siRNA-mediated CD44 targeting ($n = 6$ devices, five devices for the siRNA control). **E, F** Relationship between normalized (to total protein concentration) expression of CD44 isoforms **e** CD44s and **f** CD44v and extravasation efficiency ($n = 6$). Statistical significance was assessed by student's $t$ test assuming normally distributed data, $p < 0.05$ *, $p < 0.01$ **, $p < 0.001$ ***, $p < 0.0001$ ****.

junctions, breaching the basal membrane, and ultimately leading to migration into the stroma. The same processes, summarized in Fig. 5B, were observed for the wild-type cell line (Supplementary Fig. 5), confirming that these observations are not an artifact of CD44 overexpression. Neutralization of CD44 on MDA-MB-231 wild type cells through a blocking antibody before perfusion into the MVNs decreased extravasation efficiency 3.2-fold (Fig. 5C). Neutralizing CD44 on the endothelium had no additional effect, as co-treatment of TCs and ECs produced a similar reduction in extravasation, consistent with the absent expression of CD44 in the MVN ECs. Similarly, extravasation efficiency was reduced 1.5- or 2-fold by treatment of TCs with two siRNAs directed against CD44 (Fig. 5D). These results suggest that the reduction in metastasis formation observed in vivo after neutralization of CD44 in perfused TCs[56] could be attributed to inhibited interactions between TC CD44 and HA deposited on the endothelium and TC CD44 and other ECM components of the subendothelial ECM.

Confounding evidence exists for the role of the two most common isoforms of CD44 in determining metastatic potential[48]. CD44s, upregulated during epithelial to mesenchymal transition[57], is located on TC invadopodia where it contributes to ECM degradation and migration[56]. CD44v, however, possesses more binding motifs to the ECM[58] and it disrupts EC junctions[59]. Here, we quantified the extravasation efficiency of all TCs and compared them to the normalized expression of both CD44 isoforms. The MCF7 cells had expression levels of CD44s and CD44v near zero and the lowest extravasation efficiency (11%, Fig. 5E, F). For the remaining cell lines, expression of CD44s was negatively correlated with extravasation efficiency while expression of CD44v was positively correlated with extravasation efficiency (Fig. 5E, F). Of the renal carcinoma lines, SN12PM6 had the highest extravasation efficiency (48% compared to 21% for SN12C, $p < 0.0001$), consistent with its known enhanced

invasiveness[55]. Comparison with the arrest efficiency results in Fig. 4 shows that overall CD44 expression appears to determine TC arrest and adhesion to the endothelium. At the same time, the expression of CD44v strongly correlates with HA expression on the TCs assessed and their extravasation capacity, possibly pinpointing it as the critical isoform for metastasis and a possible target for therapeutic intervention to reduce extravasation of TC during hematological dissemination.

**Discussion**

We described a mechanism for firm adhesion of TCs to the microvascular endothelium through deposition of their GCX in the microvasculature. HA associated with TCs locally primes an adhesive niche by transient adhesion through CD44 binding. Remarkably, this mechanism is predominantly driven by the TCs, as EC activation is not required for HA-mediated adhesion. In the case of circulating TC clusters, the formation of an adhesive vascular niche by HA deposition from the cells that initially contact the endothelium may facilitate binding and extravasation of other TCs within the cluster. In the case of single circulating TCs, deposited HA may also promote adhesion of subsequent TCs traversing the same blood vessel, as observed in the MVNs. However, the likelihood of this may be low in vivo due to the typically small numbers of circulating TCs[12].

Arrest of TCs is followed by GCX-mediated adhesion to the subendothelial basement membrane and ECM as TCs migrate across the endothelium through CD44v binding. This ECM-mediated *trans*-endothelial migration mechanism is consistent with the previous observation made by us and others that TC integrins like integrin β1 contribute to extravasation[18,28], as the TC GCX provides enhanced clustering of integrin binding sites on TCs[60,61] and CD44 was observed to co-localize with such integrin clusters[62]. It is important to note that our experiments present limitations in that known factors affecting metastasis are

lacking, such as the presence of immune cells[63,64] and organo-typic variations in endothelial expression of receptors and consequent extravasation pattern differences[65,66]. Nevertheless, the results reported are for multiple human cancer cell lines in an entirely human cell-based vascular model that captures physiologically relevant key aspects of the human microvasculature. As such, these studies provide insight into potentially crucial GCX-mediated mechanisms that could be targeted therapeutically to hinder metastasis.

## Methods

**MVN formation, GCX removal, and permeability**. MVNs were cultured using vasculife endothelial medium (LL-0003, Lifeline) and pooled HUVECs (GFP-expressing, Angio-Proteomie, 6 million ml$^{-1}$ after five passages) and nHLFs (Lonza, 2 million mL$^{-1}$ also after five passages) in fibrin hydrogel exactly as previously described[31], including seeding of the endothelial monolayer on the sides of the hydrogel at day 4 of culture and use at day 7. The choice of these cells was dictated by their propensity to robustly form MVNs in microfluidic devices. Enzymatic removal of GCX components was achieved using hyaluronidase (H3631, Sigma Aldrich, used at 150 units mL$^{-1}$), chondroitinase (C3667, Sigma Aldrich, used at 0.5 units mL$^{-1}$), and heparinase (H8891, Sigma Aldrich, used at 0.5 units mL$^{-1}$) in vasculife, exposing either the MVNs (through perfusion) or the TCs for 10 min before washing with vasculife twice. Permeability of 70 kDa dextran (0.1 mg mL$^{-1}$, FD70, Sigma Aldrich) was measured exactly as previously described[29], using the fluorescent signal of FITC conjugated to the dextran and imaged on a Olympus FV1000 confocal microscope with custom enclosure for temperature and atmosphere control. Average vessel diameter was also extracted from the morphological analysis performed as part of the permeability measurements, as previously described[29].

**TC extravasation and arrest efficiency**. MDA-MB-231 and MCF7 cell lines were obtained from ATCC and cultured in Dulbecco's Modified Eagle's Medium (ThermoFisher) with 10% fetal bovine serum (FBS) added (ThermoFisher). SN12C and SN12PM6 cells were obtained from the MD Anderson Characterized Core Facility where they were originally isolated[67,68]. SN12C and SN12PM6 cells were maintained in Eagle's Minimum Essential Medium with 10% FBS (both Thermo-Fisher). Extravasation experiments were conducted in MVNs cultured within AIM microfluidic chips (AIM Biotech, gel channel width of 1.3 mm and 250 μm height). TCs were suspended in vasculife at 1 million cells mL$^{-1}$, and 10 μL (100 cells) were perfused in the MVNs under a transient flow, with a fraction of the cells remaining in the side (media) channels of the device upstream of perfusion. After 6 h the samples were fixed and imaged on the confocal microscope using a 10× objective and z-stacks with 5 μm spacing. Analysis of extravasation efficiency was conducted as previously described[30] using the ImageJ distribution Fiji[69]. Neutralization of CD44 was achieved using a monoclonal blocking antibody (10 μg mL$^{-1}$, 08-9407-2, American Research Products) or siRNA directed against CD44 (50 μM of siRNA in 100 μL of opti-MEM medium per well of 24-well plate for 4 h; siRNA 1 is s2681, siRNA 2 is s2683, siRNA ctrl is 4390843, opti-MEM is 31985088, ThermoFisher). Arrest and permeability experiments were conducted using custom-made three-channel polydimethylsiloxane (PDMS, 4019862, Ellsworth Adhesives) microfluidic devices with large width channels (3 mm, height of 500 μm, length of 1 cm[29,31,33]). While AIM chips use triangular vertical posts to contain the fibrin gel upon injection, the larger devices achieve that with a triangular guide-edge running at the top of the device with a depth and width both of 200 μm. Cells were again suspended at 1 million cells mL$^{-1}$, and 10 μL (100 cells) were injected in one of the media channels. A Fluigent pressure regulator (FlowEz) was then connected to the channel with tumor cells with custom connectors used previously[31], and the pressure increased to the desired value. The samples were imaged on a Nikon Eclipse Ti microscope with a 4× objective. Within 5–10 min, long before the fluid in the pressurized side channel was depleted, all TCs were observed to stop moving under flow, either due to arrest in the MVNs or because of adhesion to the bottom surface of the side channels up- or downstream of perfusion. At that point, TCs in the MVNs and in the downstream channel were manually counted using a clicker. Bead speed under varying applied pressure 200–600 Pa was similarly measured, by injection of 2 μm-diameter fluorescent beads (F13083, ThermoFisher) and video recording (10 s) of their movement on the fluorescent microscope, followed by analysis on Fiji of the bead track lengths during the acquisition time set (200 ms).

**Cell migration analysis**. Analysis of cell migration was conducted on the *xy* plane using projections of z-stacks in Fiji. MDA-MB-231 wild type cells were tracked at each time point for the first 6 h or until a TC commenced extravasation using the Manual Tracking function within the Trackmate plugin. Luminal migration speed was calculated as the distance traveled within the vasculature per time point divided by the time between time points, 30 min, then averaged for all time points. For each of the three experimental conditions, all applicable tumor cells in two

ROIs from two devices were tracked. Care was taken to analyze single cells entrapped or adhered to the vessel walls, rather than tumor cell clusters.

**Protein expression and imaging**. Expression analysis of adhesion proteins was performed using a Proteinsimple Sally Sue automatic western blot machine. MVNs within large devices were treated with the enzymes or TNFα (5 ng mL$^{-1}$, 300-01A, Peprotech) and processed after 6 h. The PDMS making up the devices was separated from the glass coverslip using a razor blade, and the whole fibrin gel channel was collected and immersed for 10 min on ice in 25 μL of cell lysis buffer (fresh stock of 10 mL prepared each time from 1× buffer (9803S, Cell Signaling Technologies) with the addition of 1 μL of Benzonase Nuclease (E8263, Sigma Aldrich) and one tablet of protease inhibitor cocktail (11836170001, Sigma Aldrich)), before storing at −80 °C. The antibodies were all used at a 1:200 dilution, against E-selectin (BBA16, R&D Systems), P-selectin (701257, ThermoFisher), VCAM-1 (AB134047, Abcam), ICAM-1 (MA5407, ThermoFisher), and CD44 (GTX102111, GeneTex). The isoforms of CD44 were taken as those having peaks around 90 kDa (CD44s) and 160 kDa (CD44v). The signal was normalized to the expression of CD31 (1:200, ab32457, Abcam) for the EC adhesion molecules, or β-actin (1:200, 926-42210, Li-Cor) for the expression of ICAM-1 from HUVECS and nHLFs alone and CD44 from TCs. Normalization to total protein content was done when comparing the expression of CD44 in the different TC types. Imaging of ICAM-1 (62133S, Cell Signaling Technologies), ZO-1 (33-9100, ThermoFisher) and CD44 (antibody above) was achieved through immunostaining, also using a 1:200 dilution for all antibodies. HA and CS were stained as done previously[70], using biotynilated HA-binding protein (1:100, 385911, Sigma Aldrich, USA) and biotynilated GSLII (1:100, B-1215, Vector Laboratories), followed by an anti-biotin IgG fragment (1:200, 200-542-211, Jackson Laboratories). HS was stained using an antibody (1:200, 370255, Amsbio). Quantification of HA expression of cells in well plates was performed by imaging on a Zeiss Axiovert fluorescent microscope with a 20× objective. The signal was normalized to the DAPI signal (D1306, Thermo-Fisher) to account for varying cell number.

**Fluorescent CD44 expressing TCs**. Human CD44s was cloned into the N terminus of pHAGE-mCherry empty plasmid by Gibson assembly[71]. The resulting construct was co-transfected with psPAX2, pMD2.G for packaging into HEK293T cells and the supernatant of the HEK293T culture was collected at day 2 and 3 post-transfection, followed by Lenti-X™ Concentrator (Takara Bio) mediated concentration. TCs were incubated with the lentivirus in the presence of polybrene, 8 μg/ml, overnight. Cells were washed twice and expanded for fluorescence-activated cell sorting. CD44S pBabe-puro was a gift from Bob Weinberg (Addgene plasmid # 19127; http://n2t.net/addgene:19127; RRID: Addgene_19127[72]). Analysis of CD44 expression by western blot demonstrated over-expression of both CD44s (lower) and CD44v (higher).

**Statistics and reproducibility**. Statistical analyses were performed using the software OriginPro2016. Details of the tests used, number of replicates and their definition, and the calculated parameters reported are given in each figure caption. All underlying data are reported in Supplementary Data 1.

**Reporting summary**. Further information on research design is available in the Nature Research Reporting Summary linked to this article.

## Data availability

The datasets generated during the current study are available from the corresponding authors on reasonable request. All data underlying the graphs presented in the main figures are available as Supplementary Data 1.

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

## Acknowledgements

G.S.O. is supported by an American–Italian Cancer Foundation Post-Doctoral Research Fellowship. C.H. is supported by the Ludwig Center for Molecular Oncology Graduate Fellowship and by the National Cancer Institute (U01 CA202177). Z.W. is supported by the Ludwig Center Fund Post-Doctoral Research Fellowship. We are thankful to Mark Gillrie for help with transfection of tumor cells and Charles Knutson, Dean Hickman, and Patricia Amarante at Amgen Inc. for training and access to the Protein Simple western blot assay. We also acknowledge the Koch Institute Swanson Biotechnology Center for technical support, specifically the Flow Cytometry Core Facility.

## Author contributions

G.S.O., M.F.C., and R.D.K. designed the study; G.S.O., C.H., C.R.F., Z.W., and L.I. performed the experiments; G.S.O. wrote the first draft of the paper, and all authors contributed to its final form.

## Competing interests

The authors declare the following competing interests: R.D.K. is a co-founder of AIM Biotech that markets microfluidic systems for 3D culture. Funding support is also provided by Amgen, Biogen, and Gore. The remaining authors declare no competing interests.
