## [Peer Review File · Communications Biology]

Reviewers' comments:

Reviewer #2 (Remarks to the Author):

This manuscript describes an experimental study to examine how the endothelial glycocalyx affects the adhesion of circulating tumor cells in the vasculature. Researchers at MIT have been examining the biological function of the glycocalyx for quite some time, dating back to the works by C. Forbes Dewey. One intriguing theory that the authors put forward is that cancer cells might shed HA that then becomes part of the endothelial glycocalyx, thereby helping the adhesion of future cancer cells that might pass through that vessel. For this reviewer, it's hard to imagine that rare circulating tumor cells might shed HA in sufficient quantity to significantly affect cancer cell adhesion in vivo. My comment relates to the point that in a milliliter of cancer patient blood the cancer cells are outnumbered by leucocytes by a factor of 10^5 or so. Nevertheless, the authors present some data supporting this idea, that was collected in a microfluidic system designed for this purpose. The authors do get some nice capillary networks to form. The networks seem quite branched and unstructured, like what you might find in the lung or kidney. This summary contains several comments that need to be addressed, and more comments are listed below:

1. One problem I have with this work is that the authors use the term "circulating tumor cells" to refer to the immortalized cell line MDA-MB-231. This is not quite right. To be rigorous, you should only use the term CTC when referring to an actual CTC isolated from patient blood, or from an in vivo animal of tumor growth. The title, abstract, etc. should not call these cells CTCs unless they also analyzed patient blood samples.

2. Previous works have elucidated the role of CD44v as an adhesion molecule, so this part of the study is not novel.

Reviewer #3 (Remarks to the Author):

Glycocalyx-mediated vascular dissemination of circulating tumor cells. Roger D Kamm et al.

This study uses a perfusable microvascular network of HUVECs and manipulation of both the endothelial and tumour glycocalyx to examine the role of glycocalyx components on adherence and extravasation of circulating tumour cells.

This is a well written manuscript describing an elegant, scientifically sound and interesting study. There is a clear background section, comprehensive description of the methods used and in general appropriate conclusions have been drawn from the data presented. The authors have used parametric statistics i.e. student's t-test, throughout, although it is doubtful that with the sample sizes studied a normal distribution could be demonstrated - no testing for normality is reported.

I do wonder if the conclusions drawn from the data presented in figure 5 are a little too dogmatic. The data are interesting and suggestive, but not definitive, so maybe the language should be toned down.

I also recommend that slightly more detail be included in some areas.

Firstly, the authors state (correctly, in their introduction) that the role of the glycocalyx in tumour cell extravasation is not resolved. However, there is a small body of literature that does describe how the endothelial glycocalyx might influence this process and I recommend that at least some of this literature is discussed.

Secondly, the authors have used HUVECs for these assays which are large vessel, foetal cells. Although the authors do very briefly mention organotypic endothelial variations in their discussion it is not clear to me why they chose to study HUVECs vs microvascular endothelial cells from an organ that does at least suffer from metastatic spread. A brief justification of the choice of endothelial cells is recommended.

Reviewer #2 (Remarks to the Author):

This manuscript describes an experimental study to examine how the endothelial glycocalyx affects the adhesion of circulating tumor cells in the vasculature. Researchers at MIT have been examining the biological function of the glycocalyx for quite some time, dating back to the works by C. Forbes Dewey. One intriguing theory that the authors put forward is that cancer cells might shed HA that then becomes part of the endothelial glycocalyx, thereby helping the adhesion of future cancer cells that might pass through that vessel. For this reviewer, it's hard to imagine that rare circulating tumor cells might shed HA in sufficient quantity to significantly affect cancer cell adhesion *in vivo*. My comment relates to the point that in a milliliter of cancer patient blood the cancer cells are outnumbered by leucocytes by a factor of 10^5 or so. Nevertheless, the authors present some data supporting this idea, that was collected in a microfluidic system designed for this purpose.

We thank the reviewer for their comments. We agree that the mechanism by which HA shed by passing TCs may help subsequent TCs arrest is speculative given the low likelihood of multiple TCs traversing the same blood vessel *in vivo* compared to the microvascular networks used here. As a first change, the language associated with this mechanism in the Results has now been modified to express a more speculative tone (lines 221-225 and the caption for Figure 4).

We also understand that some confusion may have been created by this language regarding our description of the results on the role of HA in TC adhesion to the endothelium. It was our intention to differentiate between the TCs directly depositing HA, versus subsequent TCs binding to HA previously deposited on the endothelium. As expressed above, we agree with the reviewer that the latter is less likely to play a role *in vivo*, although it could be an interesting mechanism where cells from the primary tumor are priming a distant site for subsequent colonization. The former, however, is the primary finding of the study and we believe that adequate evidence was presented to show that TC-associated HA is critical for their adhesion to the endothelium. To direct the reader, we have extended the Discussion (lines 313-323) to better convey this distinction and the important point raised by the reviewer about the likelihood of subsequent TC arrests.

As an aside, we were pleased to see the recognition of Forbes Dewey's pioneering role in identifying the important role of the glycocalyx; the senior author was supervised by Prof Dewey at the beginning of his career in the 1970's, and has collaborated with him as recently as this year through co-supervision of a doctoral student.

The authors do get some nice capillary networks to form. The networks seem quite branched and unstructured, like what you might find in the lung or kidney. The networks form connected lumens in 3D that branch several times along the paths connecting the side channels of the device. We have previously quantified network morphology showing significant similarities in vessel diameter, specific surface area, and volume fraction to microvasculature *in vivo* (ref. 29). Local functional features of the networks like expression of tight junctions, deposition of basement membrane, and expression of a glycocalyx support their physiological relevance as model systems to study TC extravasation (refs. 5, 28, 37). We appreciate the reviewer's suggestion that the networks may recapitulate organ-specific vascular morphologies, which we have not yet assessed.

This summary contains several comments that need to be addressed, and more comments are listed below:

1. One problem I have with this work is that the authors use the term "circulating tumor cells" to refer to the immortalized cell line MDA-MB-231. This is not quite right. To be rigorous, you should only use the term CTC when referring to an actual CTC isolated from patient blood, or from an in

vivo animal of tumor growth. The title, abstract, etc. should not call these cells CTCs unless they also analyzed patient blood samples. The term “circulating tumor cell” was only used in the title of the manuscript, which we have now changed to better convey the findings of this study. We agree with the reviewer that the MDA-MB-231 cell line is a simple model for patient circulating tumor cells, although it is derived from tumor cells originating from a metastatic site. Its choice was dictated by practical reasons, as stated on line 70 of the manuscript: “MDA-MB-231 breast cancer cells were chosen as model disseminating TCs due to their well-documented ability to rapidly extravasate from the MVNs^{28,37}”.

2. Previous works have elucidated the role of CD44v as an adhesion molecule, so this part of the study is not novel. The adhesive role of CD44 has indeed been previously elucidated (refs. 48,58,59). The current study provides new insight into the role of CD44 in microvascular arrest and transmigration. In particular, we show, for the first time, images of CD44 anchors connecting tumor cells to HA deposited on the endothelium. We also show that CD44-HA interactions are not needed for arrest of TCs in small vessels, but they are required for successful TC adhesion and extravasation. Finally, we assessed the relative contributions of the isoforms CD44s and CD44v during extravasation of TCs, finding that CD44v is the critical driver for the process, thus shedding light on this contested issue (see lines 279-282 of the manuscript).

Reviewer #3 (Remarks to the Author):

This study uses a perfusable microvascular network of HUVECs and manipulation of both the endothelial and tumour glycocalyx to examine the role of glycocalyx components on adherence and extravasation of circulating tumour cells. This is a well written manuscript describing an elegant, scientifically sound and interesting study. There is a clear background section, comprehensive description of the methods used and in general appropriate conclusions have been drawn from the data presented. We thank the reviewer for their comments. We did find a few typos in the manuscript, which have now been corrected.

The authors have used parametric statistics i.e. student's t-test, throughout, although it is doubtful that with the sample sizes studied a normal distribution could be demonstrated - no testing for normality is reported. As the reviewer suggests, the sample sizes assessed are, in most cases, too small ($3 < n < 6$) to demonstrate a normal distribution. For those cases, we have now added details of the assumption of a normal distribution to each figure caption. We could assess normality for the bead speeds (Figure 2b,d, using the Kolmogorov-Smirnov test), for which we have now added details in the figure caption.

I do wonder if the conclusions drawn from the data presented in figure 5 are a little too dogmatic. The data are interesting and suggestive, but not definitive, so maybe the language should be toned down. We have modified the language regarding Figure 5 (lines 295-297 of the manuscript) to better convey that the trends we observed are only for the TCs we assessed.

I also recommend that slightly more detail be included in some areas. Firstly, the authors state (correctly, in their introduction) that the role of the glycocalyx in tumour cell extravasation is not resolved. However, there is a small body of literature that does describe how the endothelial glycocalyx might influence this process and I recommend that at least some of this literature is discussed. We agree with the reviewer that a very few studies have investigated the role of the endothelial glycocalyx in extravasation. Our reading of the literature revealed studies suggesting that the endothelial glycocalyx acts as a repulsive barrier to adhesion by tumor cells (lines 20-24, refs. 19-

23), a proposed mechanism where the endothelial glycocalyx may actively bind to HA on tumor cells (lines 212-213, ref. 51), and that tumor-secreted pro-inflammatory factors may result in shedding of the endothelial glycocalyx (lines 159-160, ref. 16). Additional speculation for the role for the endothelium in extravasation appears in review articles ref. 20 and 24. Our review of the literature did not reveal additional relevant references.

Secondly, the authors have used HUVECs for these assays which are large vessel, foetal cells. Although the authors do very briefly mention organotypic endothelial variations in their discussion it is not clear to me why they chose to study HUVECs vs microvascular endothelial cells from an organ that does at least suffer from metastatic spread. A brief justification of the choice of endothelial cells is recommended. HUVECs have become the preferred model microvasculature because of their broad availability and ability to robustly self-assemble into lumenized networks (see, for example, ref. 30). We showed in the past that HUVEC networks exhibit relevant microvascular geometry and functionality. We have also shown here that networks made from HUVECs can be activated by pro-inflammatory factors in a pathophysiological way (Figure 3). As such, HUVECs reproduce key endothelial responses to tumor cells that are relevant for the present study. We have added a note to justify the use of HUVECs for this study (line 341).

REVIEWERS' COMMENTS:

Reviewer #2 (Remarks to the Author):

The authors have done a good job of responding to my previous comments. I like this study and think that it is suitable for publication in its current form.